# Effect of the Equal Channel Angular Pressing on the Microstructure and Phase Composition of a 7xxx Series Al-Zn-Mg-Zr Alloy

**DOI:** 10.3390/ma16196593

**Published:** 2023-10-07

**Authors:** Anwar Qasim Ahmed, Dániel Olasz, Elena V. Bobruk, Ruslan Z. Valiev, Nguyen Q. Chinh

**Affiliations:** 1Department of Materials Physics, Eötvös Loránd University, Pázmány Péter Sétány 1/A, 1117 Budapest, Hungary; anwar.ahmed.qasim@gmail.com (A.Q.A.); olasz.dani96@gmail.com (D.O.); 2College of Science, University of Kufa, Najaf 54001, Iraq; 3Institute for Technical Physics and Materials Science, Centre for Energy Research, Budapest Konkoly-Thege út 29-33, 1121 Budapest, Hungary; 4Institute of Physics of Advanced Materials, Ufa University of Science and Technology, 32 Zaki Validi Str., 450076 Ufa, Russia; e-bobruk@yandex.ru (E.V.B.); ruslan.valiev@ugatu.su (R.Z.V.); 5Laboratory for Dynamics and Extreme Performance of Advanced Nanostructured Materials, Saint Petersburg State University, 199034 St. Petersburg, Russia

**Keywords:** AlZnMg alloy, ECAP, UFG, hardness, precipitates, DSC, specific enthalpy, activation energy

## Abstract

A supersaturated Al-4.8%Zn-1.2%Mg-0.14%Zr (wt%) alloy was processed by the equal-channel angular pressing (ECAP) technique at room temperature in order to obtain an ultrafine-grained (UFG) microstructure having an average grain size of about 260 nm. The hardness and microstructural characteristics, such as the phase composition and precipitations of this UFG microstructure were studied using depth-sensing indentation (DSI), transmission electron microscopy (TEM), as well as non-isothermal scanning of differential scanning calorimetry (DSC), and compared to the properties of the un-deformed sample. Emphasis was placed on the effect of the UFG microstructure on the subsequent thermal processes in DSC measurements. It has been shown that the ECAP process resulted in not only an ultrafine-grained but also a strongly precipitated microstructure, leading to a hardness (2115 MPa) two and a half times higher than the initial hardness of the freshly quenched sample. Because of the significant changes in microstructure, ECAP has also a strong effect on the dissolution (endothermic) and precipitation (exothermic) processes during DSC measurements, where the dissolution and precipitation processes were quantitatively characterized by using experimentally determined specific enthalpies, ΔH and activation energies, Q.

## 1. Introduction

The Al-Zn-Mg composition (7xxx series) age-hardenable alloys have been considered asthe most important basic materials in the aluminum industry. These alloys are widely studied due to their technological and practical importance in the manufacture of automobiles, aircraft, and other construction materials [1,2,3,4,5]. It is well-established that the 7xxx materials have high mechanical properties [6,7,8,9,10] due to the fact that their microstructure can be changed on a very wide scale from the supersaturated solid solution state by applying different heat treatments [6,7] and/or severe plastic deformation techniques [11,12]. In general, if the supersaturated alloy is artificially aged under different conditions, various metastable and stable precipitates can be formed. In most cases, the change of both microstructure and mechanical properties in supersaturated AlZnMg alloys starts with natural aging at room temperature (RT), immediately after solution treatment and quenching. The decomposition of the supersaturated solid solution near RT takes place by the formation of Guinier-Preston (GP) zones. At somewhat higher aging temperatures, about between 80 and 150 °C, metastable intermediate η′ phase particles can be formed directly from the supersaturated solid solution state and/or mediately from the GP zones. Both GP zones and η′ phase particles have significant strengthening effects on the mechanical properties of 7xxx alloys. The formation of stable (equilibrium) η phase precipitates with a composition of MgZn_2_ can be expected at higher temperatures in the case of conventional AlZnMg alloys, where the history of the sample, such as crystal defects—vacancies, dislocations, grain boundaries…etc.—introduced by pre-deformation, may play an important role in the subsequent precipitation processes.

When a pre-deformation becomes severe plastic deformation (SPD), both the grain structure and precipitate structure of an AlZnMg alloy may change significantly [13]. In the last twenty-five years, the SPD technique using equal channel angular pressing (ECAP) was widely applied to refine the average grain size, resulting in an ultrafine-grained (UFG) microstructure, improving the mechanical properties of metal alloys via the Hall–Petch effect [11,12,14,15,16]. Also, some recent works show that the precipitation of dispersed second phases in the form of interlayers or segregations at grain boundaries and in grain interiors plays an important role in achieving simultaneously strength, ductility, stress corrosion, fatigue, superplasticity at lower temperatures, and thermal and deformation stability in UFG or nanostructured Al alloys [17,18,19,20,21,22,23,24]. For example, it has been shown [25] that additional alloying with Zr contributes to the formation of the Al_3_Zr particles coherent to the aluminum matrix, which causes an additional strengthening effect and actively hinders recrystallization.

Although the microstructures of several Al alloys processed by ECAP have been extensively investigated in the literature, the effect of ECAP processing on the structural and phase transformations, as well as on the subsequent change of mechanical properties in Al-Zn-Mg alloys of the 7xxx series has not been clarified in detail.

In this work, the effect of the ECAP process on the phase composition and microstructural properties of an Al-4.8%Zn-1.2%Mg-0.14%Zr (wt.%) composition alloy was studied by using depth-sensing indentation (DSI), transmission electron microscopy (TEM), and differential scanning calorimetry (DSC). The focus was also on the effect of the UFG microstructure on the subsequent thermal processes in DSC measurements, where the relevant endothermic and exothermic processes were analyzed both qualitatively and quantitatively.

## 2. Materials and Methods

A ternary aluminum alloy of chemical composition Al-4.8Zn-1.2Mg-0.14Zr (wt.%) was treated via casting procedure. This alloy is one of the most popular Al alloys of the 7xxx series and is of increased interest to the industry. The as-cast material was homogenized by heat treatment in the air for eight hours at 743 K and then hot extruded to a sheet shape of 10 × 50 mm^2^ cross section at 653 K. The extruded material alloy is characterized by a relatively large grain size of 10 μm. Billets of cylindrical shape are manufactured from extruded sheets with dimensions of 10 mm and 70 mm in diameter and length, respectively. These billets were imposed to solution heat treatment at 743 K for half an hour and quenched into room temperature (RT) water to form a supersaturated solid solution. This state is denoted as Q (quenched). The billets were then processed by the ECAP technique at RT, in four passes, following route Bc in which the processed billets are rotated in the same sense around their longitudinal axes by 90° after each pass. The ECAP die had an internal channel angle of 90°. ECAP die of the internal channel of the right angle and an outer arc of curvature at the intersection of the two parts of the channel of 20° work to impose a strain of 1 on each separate pass to induce grain refinement and to form a higher dislocation density [26,27]. In order to avoid break due to the strengthening effect of Guinier–Preston (GP) zones [28], the ECAP process was started within 10 min following the quenching. More details about the sample can be found in a previous work [13]. In addition, a quenched sample was only stored (naturally aged) at room temperature (RT) and followed by indentation measurements. In order to obtain definitive results, the microstructure was examined only in the case of ECAP-processed and only quenched samples that had been stored at RT for a long time (more than a year). These samples were marked as un-deformed ones.

Mechanical properties of the naturally-aged and ECAP-processed samples were investigated by depth-sensing indentation (DSI) measurements using a UMIS-type instrument. Indentations were carried out with a Vickers indenter tip for maximum loads of 50 mN. Together with the analysis of the indentation curves, the hardness values were evaluated using the well-known Oliver–Pharr method [29,30].

Characteristics of the microstructures, such as the matrix grains and the precipitates were characterized by transmission electron microscopy (TEM), using Titan Themis G2 200 scanning TEM (STEM) equipment. The TEM and energy-disperse X-ray spectroscopy (EDS) examinations were performed at different magnifications and techniques such as high-resolution TEM (HRTEM) and high-angle annular dark-field (HAADF) imaging mod. More details about the equipment and sample preparation can be found in the previous work [13].

Thermal analyses were performed by using Differential Scanning Calorimetry (DSC) equipment to identify the thermal events in the microstructure of the un-deformed and ECAP-processed samples. The heat effects accompanying the microstructure changes of the samples were followed by heating at different heating rates of 10, 20, 30, and 40 K/min in a Perkin-Elmer DSC2 differential scanning calorimeter (DSC) in the temperature range between 300 and 700 K, using samples of about 30 mg.

## 3. Results

### 3.1. Mechanical and Microstructural Investigations

#### 3.1.1. The Background: Hardness Behavior

In order to study the effect of ECAP on the behaviors of the investigated AlZnMg alloy, let us see first the mechanical properties of this alloy, as the background. Figure 1 shows the hardness values obtained by indentation on both the ECAP-processed and un-deformed samples (see Figure 1a), as well as some typical indentation curves (Figure 1b). Figure 1a clearly shows the well-known strengthening phenomenon of supersaturated solid solution Al-Zn-Mg alloys [7,31] during storing at room temperature (RT), the well-established strengthening effect of natural aging. Because of the fast formation of Guinier–Preston (GP) zones [7], the Vickers hardness (HV) of the un-deformed sample visibly increased in the earlier 24 h of natural aging, from 830 to 1150 MPa. After that, although at a lower rate, the hardness continues to increase to about 1500 MPa during the first month of storage at room temperature. After this growth range, the hardness apparently remains constant, but, in fact, it increases very slowly, reaching the hardness value of about 1700 MPa after one year of storing at RT. As a consequence of the strengthening effect of GP zones, the hardness of the un-deformed sample doubles during one-year storage at RT.

Experimental results also clearly show the strengthening effect of the severe plastic deformation using ECAP. The hardness of the ECAP-processed sample is 2115 MPa (see Figure 1a), which is 25% higher than the mentioned hardness of the one-year stored sample, and is almost two and half times higher than the initial hardness (830 MPa) of the freshly quenched sample.

The effect of the formation of GP zones during the early stage of natural aging can also be followed by analyzing the development of the indentation depth–load curves obtained on the un-deformed samples (see Figure 1b) [31]. It can be seen that in the very early stage—about in the first 3 h—of natural aging, characteristic indentation steps appear in the depth–load curves, indicating an intermittent indentation process in this stage of natural aging. This phenomenon corresponds to the Portevin-Le Chatelier (PLC) type plastic instabilities [32,33], or serrated yielding often observed in tensile tests. The appearance of step-like indentation reflects a significant interaction between mobile dislocations and solute atoms in the earlier stage of storing at RT after quenching. Because of the rapid formation and growth of GP zones, indicated by the increasing hardness, the interaction of dislocations with GP zones becomes dominant relative to that with solute atoms, leading to the irregular development and eventually to the disappearance of the instability load–depth steps, after about 3 h of storing at RT. From then, the indentation depth–load curves become smooth, and the strengthening effect of the GP zones appears in such a way that the maximum indentation depth decreases as the aging time increases. The indentation depth–load curve obtained on the ECAP-processed sample is also plotted in Figure 1b. It can be seen that the smallest maximum depth was observed on this indentation curve, which actually corresponds to the maximum hardness shown in Figure 1a.

#### 3.1.2. TEM Investigations

Figure 2 shows the microstructure of the initial un-deformed AlZnMgZr sample, which was only quenched and then stored at room temperature (RT) for a long time. Only Al_3_Zr particles having a diameter of 10–20 nm and finely-distributed GP zones with a size of 1–3 nm can be observed in this state of the alloy. It is well-known that Al_3_Zr particles formed during the solidification of the cast samples [34], having a grain-refinement effect. The formation of GP zones at room temperature indicated clearly the supersaturation in the composition of the sample. It can be seen that in the only quenched, but un-deformed state, room temperature is too low for the formation of other precipitates.

The ECAP processing significantly changed the microstructure of the quenched sample, resulting in the aforementioned large increase in the hardness of the alloy. As was shown previously [13], ECAP processing leads to the formation of an ultrafine-grained (UFG) structure with a grain size of about 260 nm and nano-sized second-phase precipitates. Figure 3 shows the microstructure of the ECAP-processed sample [13]. It can also be seen in Figure 3a that the ECAP-processed UFG structure contains precipitates inside both the interiors of grains and grain boundaries, as illustrated by bright areas in the TEM image. In addition to GP zones and Al_3_Zr particles, small η′- and larger η-phase MgZn_2_ precipitates can also be observed in the ECAP-processed structure, as shown in Figure 3b. The corresponding energy-disperse X-ray spectroscopy (EDS) elemental maps for Al, Zn, Mg, and Zr can be seen in Figure 3c–f, respectively.

### 3.2. Characterization of Microstructure by DSC Investigations

Figure 4 shows the DSC thermogram profiles obtained at different (V) heating rates for both un-deformed and ECAP-processed samples. In all cases, three peaks were distinguished in these profiles. The first endothermic peak observed at the lowest temperature indicates the reversion of some kind of existing phases in the sample. At the intermediate temperature of the profile, an exothermic peak was dominant, related to the precipitation and coarsening of hardening precipitates. At a higher temperature range, another endothermic peak can also be observed, denoting the dissolution process of precipitates formed through heat treatment [10,35]. From the point of view of the effect of the ECAP process, we were primarily interested in the development of the first endothermic and exothermic peaks.

In the case of the un-deformed sample (see Figure 4a), the first endothermic peak was located in the low-temperature interval of 380–438 K with a peak at 422 K when subjected to a heating rate, V of 10 K/min. This peak is probably attributed to the dissolution of Guinier–Preston (GP) zones and vacancy-rich solute clusters (VRC) [35,36]. The maximum rate of the dissolution process was identified at the peak temperature of 422 K, which shifted towards a higher temperature at 450 K and became more intense as the heating rate increased to 40 K/min. It can be seen that in the case of the ECAP-processed samples (see Figure 4b), the corresponding endothermic peak obtained for a given heating rate was located at a lower temperature compared with that of the un-deformed sample. For instance, this peak shifted to a lower temperature of 440 K when the ECAP-processed sample was measured at the heating rate of 40 K/min.

Similarly to the situation of the first endothermic one, the location of the following exothermic peaks obtained for the ECAP-processed sample also shifted to somewhat lower temperatures at every heating rate. For instance, when applying the heating rate of 10 K/min, the exothermic peak of the un-deformed sample manifested with a maximum at 504 K and extended in the interval of 465–525 K. Meanwhile, in the case of the ECAP-processed sample, the corresponding peak appeared with a maximum at 484 K and extended over the temperature interval of 437–509 K. As the exothermic peak is ascribed to the formation of η phase particles and their coarsening [37,38], the present results clearly confirm the possibility of the ECAP process to induce the formation of the η′ and η precipitates. This is also demonstrated by the larger areas belonging to the peak under the DSC profile of the ECAP-processed sample, compared with that of the un-deformed sample (see Figure 4b). 

#### 3.2.1. Specific Enthalpies Characterizing the Dissolution and Precipitation Reactions 

Analyzing the DSC thermograms, the specific heats—enthalpies—of dissolution, ∆*H_d_* and that of precipitation, ∆*H_p_* can be determined. Figure 5 shows these enthalpies obtained for the first endothermic peak (Figure 5a) and for the exothermic peak (Figure 5b) as the function of the heating rates in the case of both un-deformed and ECAP-processed samples.

Experimental results show that as the effect of the ECAP process, while the specific enthalpy obtained for dissolution, ∆*H_d_* decreased (from a range of 5.0 to 6.2 to a range of 4.2 to 3.5 J/g, see Figure 5a), the value obtained for the precipitation, ∆*H_p_* increased (from a range of 5.9 to 7.0 to a range of 4.1 to 2.6 J/g, see Figure 5b).

#### 3.2.2. Kinetic Parameters for Dissolution and Precipitation Reactions

In general, the dissolution and precipitation processes taking place in multi-phase material when imposed to DSC of non-isothermal heat treatment are described by the kinetics parameters incorporated by the Avrami–Johnson–Mehl theory [39,40,41]. These parameters include the transformed volume fraction, *Y* that can be given as: (1)$$Y\left(T\right)=\frac{A\left(T\right)}{A(T_{f})},$$
and the rate of transformation, *dY*/*dt* given by:(2)$$\frac{dY}{dt}=\frac{dY}{dT}\cdot \frac{dT}{dt}=\frac{dY}{dT}\cdot V,$$
at heating rate *V*. Here, A(T) refers to the area under the given—dissolution or precipitation—peak in the range of temperatures extended from the initial temperature, $T_{i}$ to T, and *T_f_* is the final temperature of the investigated peak in the DSC thermograms. 

Figure 6 shows the experimentally determined Y-T plots characterizing the dissolution and precipitation in both the un-deformed and ECAP-processed samples measured at different heating rates. It can be seen that these (Y-T) plots can be described typically by a sigmodal shape function and shifted to a higher temperature as the heating rate increased. The effect of the ECAP process is clearly visible on the curves. While the interval of the temperature range for the endothermic reaction in the un-deformed sample is extended over about 90 K (from 397 to 488 K) (see Figure 6a), this value is about one and a half times lower for the ECAP-processed sample, only 60 K, as it extended from 395 to 455 K (see Figure 6c). At the same time, the precipitation process (exothermic reaction) of the ECAP-processed sample is also shifted to a lower temperature range from 460 to 550 K (see Figure 6d), relative to that (from 485 to 570 K) of the un-deformed sample (see Figure 6b).

Figure 7 presents the transformation rate, *dY/dt* in the function of temperature, T at different heating rates for both the un-deformed and ECAP-processed samples. In every case, the transformation rate is low at the beginning and the end of the transformation but there is a rapid increment in between i.e., a maximum located in the middle region. It can be seen that the effect of the ECAP process is also clearly visible in this case. Considering the dissolution process (endothermic reaction), while the maximum transformation rates for the un-deformed sample are located between 425 and 460 K, the maximum rates observed in the ECAP-processed sample can be found in a narrower temperature range, from 425 to 440 K. 

#### 3.2.3. The Activation Energy of Dissolution and Precipitation Processes

Increasing the heating rate, the peak temperature, and *T_p_* of both the endothermic and exothermic peaks is shifted towards higher temperature. This behavior of the DSC results indicates that the preceding processes of dissolution and precipitates are sensitive to the heating rate, so it is thermally activated and temperature dependent. Normally the precipitates like GP zones and other strengthening phases in aluminum alloys have a composition and crystal structure different from the surrounding of the Al matrix, hence their development leads to a change in structure, and this requires a long-range diffusion which can be characterized by activation energy, *Q*. The activation energy value for both dissolution (reaction) and precipitation processes (endothermic and exothermic reactions, respectively) can be obtained by applying the Kissinger equation [40]:(3)$$\ln\left(\frac{V}{T_{p}^{2}}\right)=C+\frac{Q}{RTp},$$
where *V* is the heating rate (in K/min) and *T_p_* is the peak temperature of the given process, *C* is a material constant and *R* is the universal gas constant. The activation energy, *Q* will be denoted as *Q_d_* for the dissolution, and as *Q_p_* for the precipitation processes.

Figure 8 shows the Kissinger plots obtained for the first dissolution (endothermic) and precipitation (exothermic) processes in both the un-deformed (Figure 8a) and ECAP-processed (Figure 8b) samples. It can be seen in all cases that the data points can be fitted well with a linear line. From the slope of this fitted line, according to Equation (3) the activation energy, *Q* can be determined. Considering the experimentally obtained activation energy values, the effect of the ECAP process is also clearly visible, as this effect increases the activation energy in the case of dissolution processes but decreases it in the case of precipitation processes. While in the case of the un-deformed sample, the value of *Q_d_* is only 85 kJ/mole (see Figure 8a), because of the ECAP process it became higher, 111 kJ/mole (Figure 8b). Furthermore, the *Q_p_* value of 147 kJ/mole in the un-deformed sample decreased to 118 kJ/mole when the sample was processed by ECAP.

The experimentally determined average specific heat enthalpy ∆*H* and activation energy *Q* for both the un-deformed and ECAP-processed samples are listed in Table 1.

## 4. Discussion

The TEM investigation reveals a significant change in the microstructure in the ECAP-processed Al-Zn-Mg-Zr samples. Besides the well-known grain-refining effect, severe plastic deformation has also resulted in the formation of η′- and η-phase MgZn_2_ precipitates at room temperature. It should be noted that these precipitates are formed only at temperatures higher than 100 °C in the conventional (un-deformed) 7xxx Al alloys, where the peak hardness can be obtained when both GP zones and η′-phase particles are formed [36]. In the present case, the collective strengthening effect of the Hall-Petch mechanism [24,42,43], as well as of GP zones and MgZn_2_ particles in the ECAP-processed sample increased the hardness of the initial—freshly quenched—sample by two and a half times, from 830 to 2115 MPa (see Figure 1a).

In the metallurgy field, the specific enthalpies of dissolution, ∆*H_d_* and precipitations, ∆*H_p_* represent a fingerprint of the reversion and precipitation reactions, respectively [44,45]. Present experimental results have shown that while ∆*H_d_* is decreasing, ∆*H_p_* is increasing in the ECAP-processed sample, relative to the corresponding specific enthalpies of the un-deformed sample (see Table 1). Both tendencies confirm the existence of η′-phase particles in the ECAP sample. These particles—being more stable than GP zones—did not dissolute into the matrix during the first endothermic reactions, but could directly transform into η-phase precipitates during the subsequent exothermic reaction, causing the decrease in ∆*H_d_* and also the increase in ∆*H_p_*. This result is in full agreement with the mentioned TEM results. It should be noted that the increase in specific enthalpy of the exothermic reaction is a feature of the deformed samples because a higher dislocation density and more grain boundaries were formed. Therefore, stronger processes of precipitates are associated with ECAP-processed samples. For the exothermic reaction, the grain boundaries which work as heterogeneous nucleation sites, can facilitate the nucleation by presenting surfaces of lower inter-phase boundary energy in comparison with the interphase of precipitate and the parent Al-matrix of the un-deformed sample [37,46].

It should be noted that considering the thermally activated—time-dependent—processes in the exothermic reaction of both investigated samples, the low heating rate (10 K/min) allows more time for the formation processes i.e., there is sufficient time for the germ precipitations to convert and form the η′- and/or η-phase precipitates. This is demonstrated as a sufficient yield of the process achieved at a lower heating rate. While at a higher heating rate, the time is not enough for the germ nucleation to precipitate these phases, so a transformation of the metastable phase to a stable phase η is more likely to form and this is represented at higher temperatures on the DSC profile [38,47]. Hence, a higher heating rate induces stable precipitates of larger size and indicates that the formation of the metastable phase is more difficult due to the reversion of GP zones. These findings are coherent with the results of several previous investigations [48,49].

Analysis of the DSC results has also shown that the values of activation energies characterizing reversion *Q_d_* and precipitation *Q_p_* processes in the un-deformed sample are 84.8 and 147.4 kJ/mole, respectively, and the corresponding ones obtained for ECAP-processed samples are 110.9 and 117.7 kJ/mole (see Figure 8 or Table 1). The activation energy *Q_d_* of dissolution is smaller in the case of the un-deformed sample because this sample contained mainly GP zones, which are dissolved more easily compared with precipitates in the ECAP-processed sample. Meanwhile, in the exothermic reaction, a drop in activation energy (*Q_p_*) occurred in the ECAP-deformed sample due to high dislocation density and more grain boundaries, which work as sites for nucleation and growth of precipitates, facilitating and making the precipitation process easier. Typically, solute segregation leads to modifying the chemistry around the dislocations, so the strain field of these dislocations may affect the diffusion by compensating the strain field related to the nucleus, hence the preferable heterogeneous nucleation sites are at dislocations [37]. As the number of grain boundaries and the dislocation density are larger for the ECAP-processed sample, there is a reduction in the barrier to precipitates. This reduction is presented as the mentioned drop of activation energy, *Q_p_* from 147.4 kJ/mol to 117.7 kJ/mol for the ECAP-processed sample compared to the un-deformed sample, as demonstrated in Figure 8. Although further investigations are needed to clarify better the details of thermally activated processes—both the dissolution and precipitation—during DSC measurements, the present results are important for understanding the unusually high-plastic behavior of this UFG alloy after ECAP processing [12] which will soon be examined. On the whole, the grain-refining, strengthening, as well as precipitation-facilitating effects from equal channel angular pressing as a useful severe plastic deformation process are an exciting field in today’s materials science. 

## 5. Conclusions

The effect of the ECAP process on the hardness and microstructural properties of an AlZnMgZr alloy was studied by using DSI, TEM, and DSC techniques. The main results can be summarized as follows:It has been shown that the room-temperature severe plastic deformation exerted by ECAP resulted in not only an ultrafine-grained but also a strongly precipitated structure in the investigated AlZnMgZr alloy;As a collective consequence of grain-size (Hall–Petch) strengthening and precipitate-hardening, the ECAP process significantly—two and half times—increased the initial hardness (830 MPa) of the freshly quenched sample to 2115 MPa;It was demonstrated that the SPD via ECAP has a strong effect on the dissolution (endothermic) and precipitation (exothermic) processes during the DSC measurements because of the significant changes in the microstructure of the ECAP-processed samples;Results of the DSC tests have revealed that both the dissolution and precipitation processes are characterized by a specific enthalpy of dissolution, Δ*H*. As the effect of the ECAP process, while the specific enthalpy obtained for dissolution, ∆*H_d_* decreased from a range of 5.0 to 6.2 to a range of 4.2 to 3.5 J/g, the value obtained for the precipitation, ∆*H_p_* increased from a range of 5.9 to 7.0 to a range of 4.1 to 2.6 J/g;Results of the DSC tests have also revealed that both the dissolution and precipitation processes are characterized by an activation energy, *Q.* As the effect of the ECAP process, while the activation energy obtained for dissolution, *Q_d_* increased from 84.8 to 110.9 kJ/mole, the value obtained for the precipitation, *Q_p_* decreased from 147.4 to 117.7 kJ/mole.

The obtained results allow one to expect a significant change in the mechanical properties of the alloy after ECAP processing not only at RT, but also at an elevated temperature, and this could be a topic of further research.

## Figures and Tables

**Figure 1 materials-16-06593-f001:**
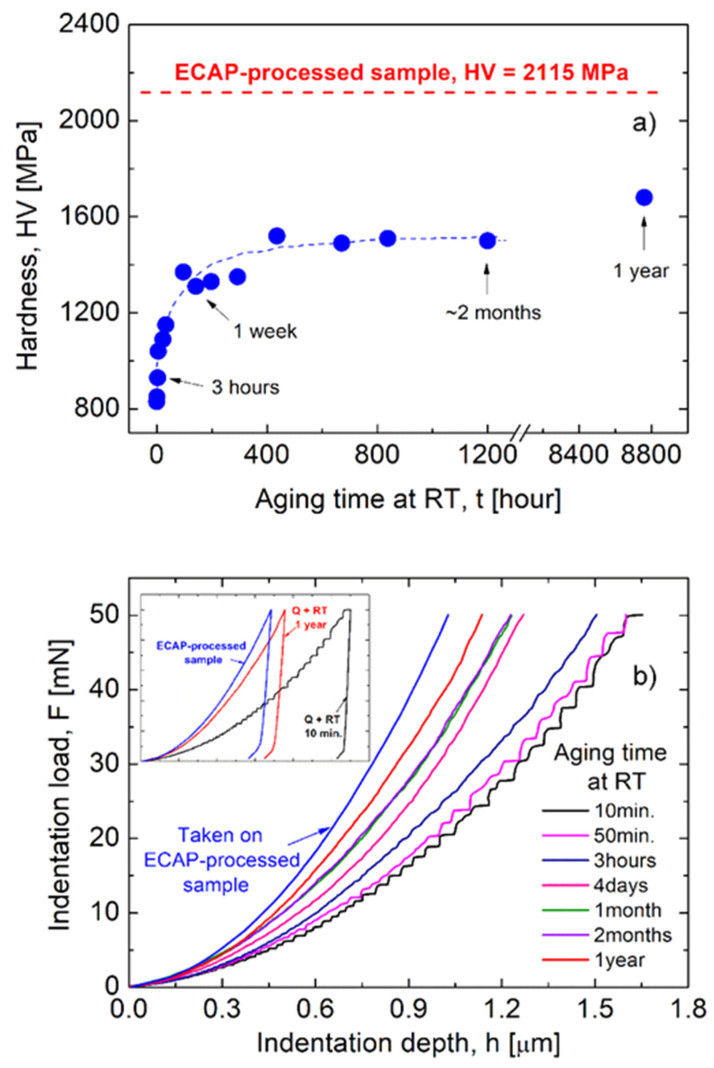
Mechanical strength of the investigated un-deformed and ECAP-processed samples as (**a**) the Vickers hardness and (**b**) the characteristic change of the indentation curves in the function of the aging time at RT.

**Figure 2 materials-16-06593-f002:**
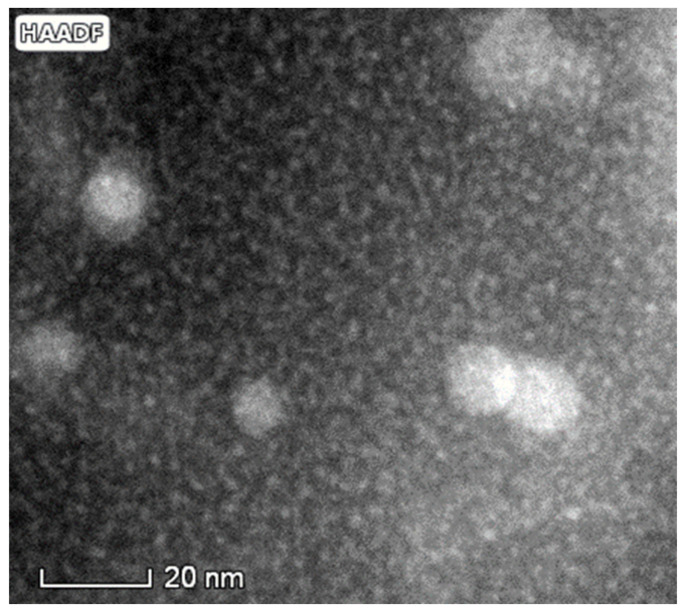
STEM HAADF micrograph of the microstructure of Al-4.8Zn-1.2Mg-0.14Zr sample after quenching and storing at room temperature for a long time (more than 1 year), showing the presence of Al_3_Zr particles and finely-distributed GP-zones.

**Figure 3 materials-16-06593-f003:**
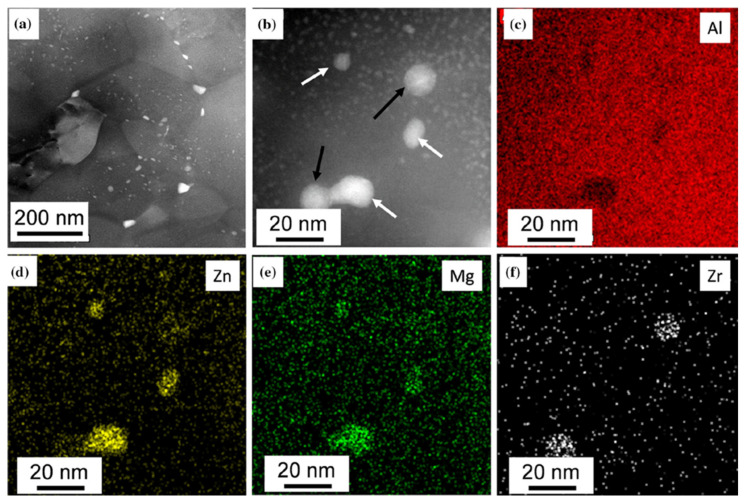
Ultrafine-grained microstructure of the investigated ECAP-processed AlZnMgZr sample taken as (**a**,**b**) HAADF STEM images in low and higher magnifications showing grains and precipitates, respectively, and (**c**–**f**) EDS elemental maps for Al, Zn, Mg, and Zr obtained on the area shown in (**b**). The white and black arrows in the image (**b**) indicate Mg/Zn- and Zr-rich large precipitates. Reproduced from Ref. [13]. Copyright 2019, Springer.

**Figure 4 materials-16-06593-f004:**
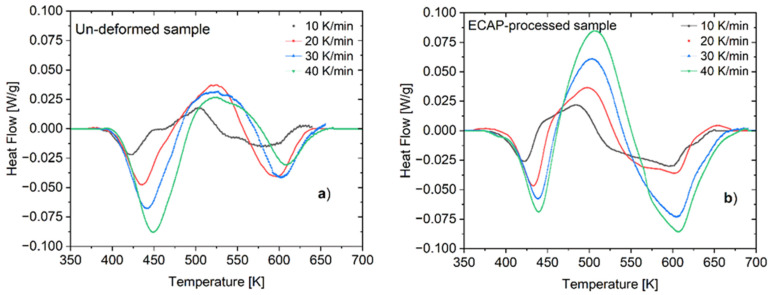
Typical DSC thermograms taken (**a**) on the un-deformed sample, and (**b**) on ECAP-processed samples at different heating rates.

**Figure 5 materials-16-06593-f005:**
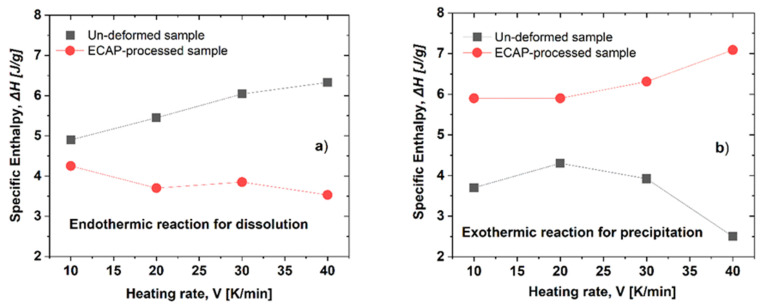
Specific enthalpy values, ∆*H_d_* and ∆*H_p_* as a function of heating rate for (**a**) dissolution (at the first endothermic peak) and (**b**) precipitation (at the exothermic peak), respectively, in both un-deformed and ECAP-processed samples.

**Figure 6 materials-16-06593-f006:**
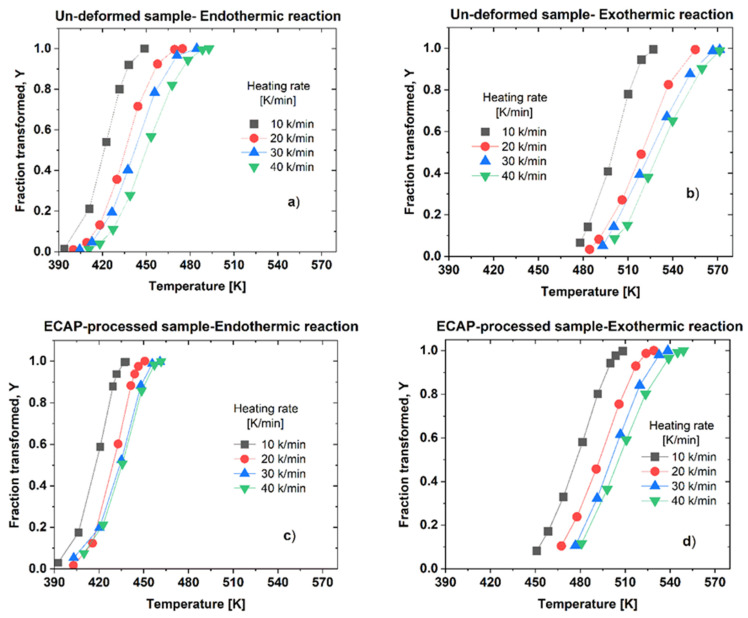
Plots of Y-T functions obtained at different heating rates, characterizing (**a**) dissolution (endothermic reaction) and (**b**) precipitation (exothermic reaction) in the un-deformed sample, as well as (**c**) dissolution and (**d**) precipitation in the ECAP-processed sample.

**Figure 7 materials-16-06593-f007:**
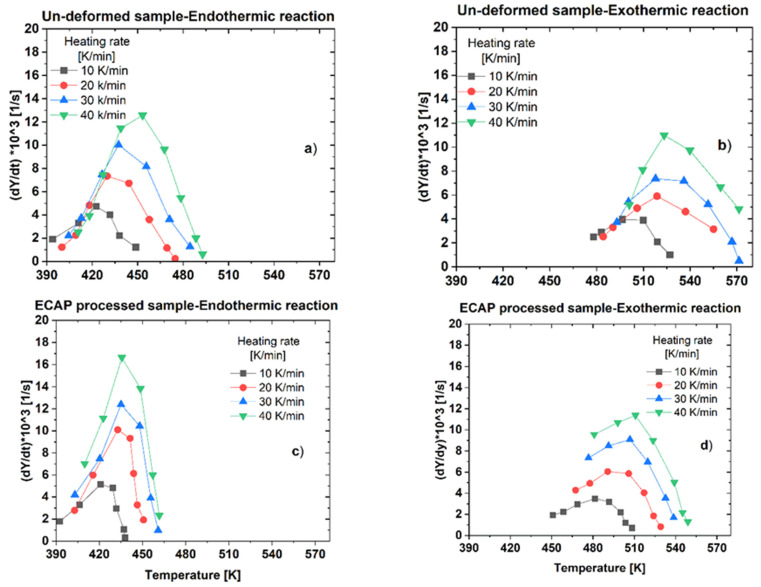
Plots of *dY*/*dt*-T functions obtained at different heating rates, characterizing (**a**) dissolution (endothermic reaction) and (**b**) precipitation (exothermic reaction) in the un-deformed sample, as well as (**c**) dissolution and (**d**) precipitation in the ECAP-processed sample.

**Figure 8 materials-16-06593-f008:**
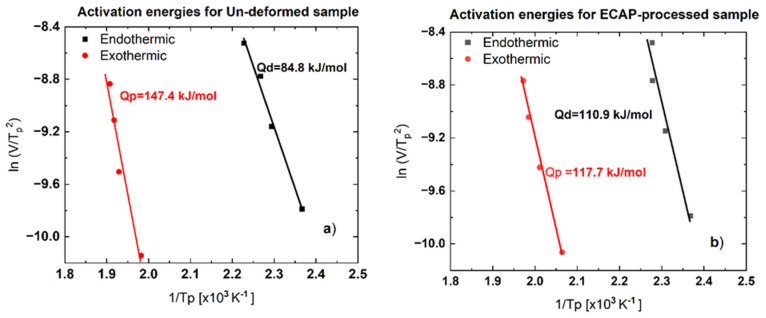
Kissinger plots for the first dissolution (endothermic) and precipitation (exothermic) processes in (**a**) un-deformed and (**b**) ECAP-processed Al-Zn-Mg-Zr samples. The relative error of calculation is lower than 5%.

**Table 1 materials-16-06593-t001:** Different types of energies obtained by analysis of the DSC measurements performed on the un-deformed and ECAP-processed Al-Zn-Mg-Zr samples. (All values were obtained within 8% relative error).

Reaction Type	Un-Deformed Sample	ECAP-Processed Sample
Dissolution (endothermic) reaction $∆H_{d}$	5.0 ÷ 6.2 J/g	4.2 ÷ 3.5 J/g
Precipitation (exothermic) reaction $∆H_{p}$	5.9 ÷ 7.0 J/g	4.1 ÷ 2.6 J/g
Dissolution reaction, *Q_d_*	84.8 kJ/mol	110.9 kJ/mol
Precipitation reaction, *Q_p_*	147.4 kJ/mol	117.7 kJ/mol

## Data Availability

The raw and processed data required to reproduce these results are available upon reasonable request.

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
