# Peer review of "Effect of the Equal Channel Angular Pressing on the Microstructure and Phase Composition of a 7xxx Series Al-Zn-Mg-Zr Alloy"

_materials, 2023, doi:10.3390/ma16196593_

Round 1
Reviewer 1 Report
The paper “Effect of the Equal Channel Angular Pressing on Microstructural and Mechanical Properties of an Al-Zn-Mg-Zr alloy” contains new useful scientific information. The work is well organized, the list of methods used is sufficient, and the results are clear. In addition, a detailed analysis of the results and a sufficient level of discussion are provided. I believe the paper can be published in “Materials”. There is only one minor remark:
Line 88. “The billets then have been processed by the ECAP technique at RT, in four 87 passes, following route Bc.”
What is Bc? Please specify.
Reviewer 2 Report
Although the authors have conducted an excellent research on the development of aluminium 7000 series alloys, there are certain weak areas that must be addressed:
1. The title should clarify that the research was conduced on AA 7xxx alloys, more specifically the designation should be mentioned for prospective readers ( like Al 7075-T6 etc.). Also economic and industrial importance of Al 7xxx alloys should be presented in terms of the total tonnage of Al 7xxx alloys as a percentage of the total tonnage of aluminium alloys in the industry. What is the significance of the 7xxx series.
2. The main concern for Al 7xxx alloys is fatigue and stress corrosion. Improvement in these properties should be reported.
3. There should be an illustrative description of the ECAP process.
4. The authors should provide a list of important mechanical properties they aim to improve as mentioned in the title and the % improvement observed in tabular form. The should include strength (yield/tensile), hardness, toughness, fatigue/endurance strength etc.
5. A list of nomenclature should be included before the introduction section.
There are minor errors that should be proofread by the authors.
Reviewer 3 Report
1. Please explain the novelty of the work.
2. Please explain the significance of fig 2. (HAADF image)
3. Please give the schematic and actual image of the equipment used in materials and methods.
4. Discussion needs to be strengthened.
5. Explain the rationale for selecting AlZnMgZr alloy.
Minor editing of English language required
Reviewer 4 Report
Dear Authors,
The layout of the manuscript is logical, the results presented are fairly analyzed. Only the list of abbreviations can be added.
Author Response
We much appreciate the detailed and helpful comments of the Reviewer.
Since this is not a long review paper and does not contain many abbreviations, we do not consider it necessary to provide such a nomenclature. Abbreviations for some very well-known, frequently occurring terms are given in the appropriate place, as in most research papers.
Round 2
Reviewer 2 Report
Although the authors have attempted to address the reviewer observations, these have not been satisfied for improved and easy readership. The authors have failed to provide the actual standard for their work material. AA 7XXX was mentioned as a generic term for the 7 series alloys. Which specific alloy did the authors use. Reference can be found in any materials handbook or even a simple search online. Also the authors have failed to provide the total tonnage of aluminium 7 series alloys as a percentage of the total tonnage of aluminum alloys in the industry.
The authors have claimed that an "illustrative description of the ECAP process" has been included. However there is no figure to support this response.
Although this article "is not a long review paper", still the number of abbreviations and acronyms used is substantial and for ease of readership and therefore improved citability a list of nomenclature should be provided. Also a clear research motivation should be mentioned. Why are the authors focused on the microstructural and phase transformations during ECAP processing. What is the benefit to the end user/industry. How will it held in improvement of material properties.
Language usage is acceptable.
